# The Numerical Investigation of the Performance of a Newly Designed Sediment Trap for Horizontal Transport Flux

**DOI:** 10.3390/s22197262

**Published:** 2022-09-25

**Authors:** Cheng Wang, Lei Guo, Shaotong Zhang, Zihang Fei, Gang Xue, Xiuqing Yang, Jiarui Zhang

**Affiliations:** 1Institute of Marine Science and Technology, Shandong University, Qingdao 266237, China; 2Key Laboratory for Submarine Geosciences and Prospecting Techniques, Ministry of Education, Institute of Estuarine and Coastal Research, College of Marine Geosciences, Ocean University of China, Qingdao 266100, China; 3College of Environmental Science and Engineering, Ocean University of China, Qingdao 266100, China

**Keywords:** sediment transport, sediment trap, flux analysis, particle capture, numerical simulation, porous media

## Abstract

Marine sediment transport is closely related to seafloor topography, material transport, marine engineering safety, etc. With a developed time-series vector observation device, the sediment capture and transport process can be observed. The structure of the capture tube and the internal filter screen can significantly affect the flow field during the actual observation, further influencing the sediment transport observation and particle capture process. This paper presents a numerical model for investigating the effect of device structure on seawater flow to study the processes of marine sediment transport observation and sediment particle capture. The model is based on the solution of both porous media and the Realizable k-ε turbulence in Fluent software. The flow velocity distribution inside and outside the capture tube with different screen pore sizes (0.300, 0.150, and 0.075 mm) is analyzed. To enhance the reliability of the numerical simulation, the simulation calculation results are compared with the test results and have good coincidence. Finally, by analyzing the motion law of sediment in the capture tube, the accurate capture of sediment particles is achieved, and the optimal capture efficiency of the sediment trap is obtained.

## 1. Introduction

Connecting terrestrial and marine environmental systems, rivers serve as the link for land–sea sediment transport and circulation. The annual sediment discharge from rivers to the ocean reaches 1.9 × 10^10^ t worldwide [1,2]. The substantial deposited particles carried by rivers into the sea are constantly transported, deposited and accumulated under the action of currents, waves and circulation, forming a large muddy sediment system in the outer shelf area [3,4]. The transport process of sediments from rivers to the sea has an important impact on estuarine geomorphological evolution, route planning and offshore engineering [5,6,7]. In addition, sediment particles act as a major carrier of marine organic matter, nutrient salts and pollutants. Their transport process affects the biological, chemical and physical marine environments of shelf seas. Therefore, studying sediment transport processes to accurately capture sediment particles is essential for gaining insight into sediment source-to-sink processes, elucidating coastal sea–land interactions, exploring global material cycles and protecting the marine ecological environment [8,9,10,11,12].

Currently, common methods used for marine sediment observation include in situ sampling, satellite remote sensing, seated-bottom observation platform and moored sediment traps [13,14]. In situ sampling can directly obtain accumulation status. However, this method requires high input costs and a long sampling and analysis period, and it cannot obtain extreme data values. Therefore, it cannot meet the refined research on observing the dynamic process of suspended matter [15]. Remote-sensing technology can macroscopically monitor land and water across the earth’s surface on a large scale but has limited judgment of bottom sediment transport, making it difficult to further investigate the causes and mechanisms of sediment transport [16,17,18,19]. The seated-bottom observation platform can indirectly estimate sediment transport by carrying acoustic and optical equipment but cannot directly capture sediment samples [20,21,22].

The transport of suspended sediments in seawater can be derived into advective transport and vertical mixed transport [23]. Moored sediment traps are mainly divided into vertical and horizontal types. The vertical sediment trap collects vertical deposited sediments from water body in time series. Then, indicators such as particle size, mineral composition and chemical composition are obtained through indoor experiments to study seafloor organic matter transport and ocean carbon sink processes [24,25,26,27,28]. Horizontal sediment traps are used to capture sediment parallel to the seafloor boundary layer for sampling and forming sediment deposition sequences. Then, physical indicators such as particle size distribution of sediments are obtained through indoor experiments to support studies on erosion and deposition in the estuary and coastal areas and long-distance sediment transport into the sea [29,30,31,32]. However, the two traps ignore the variable factors in the sediment transport process and only obtain the final sedimentation results, which seriously restricts the study of sediment transport fluxes.

Our team previously proposed a new method of observing sediment transport fluxes. A sediment transport time-series vector observation device (3-D Trap) was developed to monitor the transport process of suspended particulate matter in the seafloor boundary layer and to capture sediment particles. By combining indirect observation methods such as acoustics and optics, a multilevel sediment transport flux time-series analysis method was established based on flow velocity and suspended sediment concentration. Sediment samples at various moments were analyzed through indoor tests to obtain their physical and chemical properties. In situ long-term 3D dynamic observations of marine suspended sediment transport mechanisms were completed, revealing the spatial and temporal distribution characteristics and the controlling factors of marine sediment transport processes. Additionally, the mechanisms and characteristics of the dynamic evolution of sediment transport were quantitatively explained. The key dynamical processes of scientific research, such as erosion and deposition, material transport and element cycling, were further elucidated, providing a new perspective and technical means for scientific research on marine sediments [33].

There are shortcomings in the research on the impact of the device after placement on the seabed on the original ambient flow field and the particle capture mechanism. Therefore, based on the Fluent numerical simulation method, this paper explores the influence of factors such as the capture tube structure and the filter screen on the flow field to obtain the corresponding relationship between the flow velocity inside and outside the capture tube. In addition, the motion states of sediment particles with different particle sizes are analyzed to obtain the uplift flow velocity and stop flow velocity of different sediment particle sizes. The flow field inside the capture tube obtained from numerical simulation is combined to accurately capture sediment particles, further evaluating the efficiency of sediment capture, which is an optimized method for the trap.

## 2. Design of 3-D Trap

### 2.1. Structural Design of 3-D Trap

The 3-D trap mainly consists of a sediment capture system, observation system, control cabinet, platform frame and settlement compensation system. The bottom of the 3-D trap is a seated-bottom platform, which ensures the stability of the entire system and facilitates the capture of sediments from multidirectional transport sources on the seafloor. The overall schematic of the 3-D trap is shown in Figure 1a.

The sediment capture tube mainly includes a water flow pipe and a settling tube, as shown in Figure 1. Specifically, the front end of the water flow pipe has a horizontal water inlet, and its middle is equipped with an inclined sediment filter screen. The settling tube is fixed vertically below the screen, sealed at the bottom with a funnel-shaped opening at the top, and connected to the water flow pipe to store the captured sediment samples. The capture tube is installed with a flow meter and a turbidimeter to monitor the flow velocity and turbidity of the flow field inside the capture tube, respectively.

### 2.2. Working Principle of the Sediment Trap

The sediment trap works as follows: During the observation of the transport process of sediment particles carried by seawater on the seabed, seawater drives the transport of sediment, which flows to the inlet of the capture tube. All sediment particles with a particle size larger than the screen pore size and within the capture interval are intercepted and settled into the settling tube by gravity. Then, the water flows through the screen and out of the outlet. Thus, sediment particle samples within a precise size range can be obtained. The schematic diagram of the working principle is shown in Figure 2.

### 2.3. Analytical Equations for Sediment Transport Fluxes

The concentration *SSC_in_* and the flow velocity *V_in_* of suspended sediment can be constantly recorded during observation with the flow meter and the turbidimeter inside the device. Then, according to the correspondence between the flow velocity inside and outside the capture tube and the suspended sand concentration, the sediment transport flux of the external ambient flow field was obtained by inversion. An analytical relationship for sediment transport fluxes based on flow velocity and suspended sand concentration can be established as follows:(1)$$Q_{out}=V_{out}\times SSC_{out}$$
(2)$$V_{out}\propto V_{in}$$
(3)$$SSC_{out}\propto SSC_{in}$$

Sediment traps are deployed at seabed observation sites. During the sediment capture and the in situ observation of transport flux, the original ambient flow field will change after flowing through the capture tube, and the sediment particle capture screen inside the capture tube will also affect the flow field. Therefore, it is significant to study the relationship between flow velocity and turbidity of the flow field inside and outside the capture tube. This paper only explores the correspondence between the flow velocity inside and outside the capture tube.

## 3. Numerical Simulation of the Internal and External Flow Fields of 3-D Trap

### 3.1. Model Introduction

The physical models of porous media and Realizable k-ε provided by Fluent software were used for 3D numerical simulations and flow field simulations of the sediment trap capture process. The effects of the screen and trap structures on the ambient flow field during the in situ observation and sediment capture on the seafloor were determined.

In recent years, many scholars have used porous media models to study the fluid dynamics problems involving screen structures. They used the porous media model to simulate the filter screen and obtained the velocity distribution and hydraulic characteristics inside the filter. The feasibility of the porous media model was confirmed by comparing the numerical calculation results with the experimental results [34,35]. The Reynolds-averaged Navier–Stokes equation is the core of Fluent’s calculation of turbulent motion. Three turbulence models, namely Standard k-ε, RNG k-ε and Realizable k-ε were the result. Any turbulence model can be used together with the porous media model to simulate the flow field inside the filter. The comparison of calculation results with the physical experiment results showed that the Realizable k-ε model had the highest accuracy among the three models [36,37,38]. In addition, the porous media model has been applied to other devices with a screen structure to explore the influence of the flow field structure inside the quantitative device by the screen and to optimize the design of the device [39,40,41,42,43,44].

### 3.2. Establishment and Meshing of the 3D Model

#### 3.2.1. Establishment of the 3D Model

We established a coordinate system for the capture tube on an equal-scale model. The origin of the coordinate is at the center of the water inlet. The centerline of the inlet pipe coincides with the *X*-axis, and that of the settling tube is parallel to the *Y*-axis. The angle *α* between the screen and the wall is 45°. The structure diagram is shown in Figure 3a. For better analysis of the flow field, the *Z* = 0 m section is taken as a typical section of the capture tube cross-section.

After the 3D structure model was established, SpaceClaim (version 19.0), special pre-processing software for ANSYS Fluent, was used for flow channel extraction. Then, it was imported into ICEM CFD software to generate the mesh. To simulate the effect of the capture tube on the real flow field, the fluid domain includes the flow field inside and outside the capture tube. Porous media models were used to study the fluid dynamics problems involving screen structures. Hence, the porous media region was set up.

#### 3.2.2. Meshing

In this paper, ICEM CFD software was used to construct the capture tube mesh model, and the unstructured tetrahedral mesh was used to divide the fluid region. The capture tube mesh model is shown in Figure 3b.

To ensure the accuracy and efficiency of the calculation, the mesh in Figure 3b should be verified for mesh independence. The simulation focuses on the flow velocity and turbidity of the internal and external flow fields. Therefore, the average velocity of one section of the capture tube was taken to verify the mesh independence at the screen pore size *D* of 0.075 mm and the flow velocity outside the tube of 15 cm/s. As shown in Table 1, as the mesh number changes from 40,000 to 500,000, the change in monitoring surface velocity does not exceed 1%. It can be considered that mesh independence has been achieved. Thus, 300,000 meshes are used as the calculation meshes.

### 3.3. Setting of Parameters Related to Finite Element Simulation

#### 3.3.1. Idealized Assumptions for Numerical Calculation

For the accurate numerical study of the flow field inside and outside the capture tube, assumptions should be made before simulation calculations:

(1) Assume that seawater is a Newtonian liquid, i.e., seawater is incompressible. (2) Assume that the walls are smooth and ignore the effect of wall roughness on water transport. (3) Assume that the inlet flow of the fluid domain is uniformly distributed and does not vary with time.

#### 3.3.2. Simulation Parameter Setting

The boundary conditions and parameters of the simulation are based on typical sediment particle distribution and transport in the coastal zone. Screen mesh sizes of 50, 100 and 200 (pore size *D* = 0.300 mm, *D* = 0.150 mm and *D* = 0.075 mm) are selected for the capture tubes. The ambient flow velocities are controlled at 15 cm/s, 20 cm/s, 25 cm/s, 30 cm/s and 35 cm/s.

The simulation parameters obtained according to the actual working conditions are shown in Table 2 and Table 3.

#### 3.3.3. Porous Media Model

The porous media model in Fluent software was used to simulate the screen. The computational domain was defined in the “Porous_body” region in Figure 3b. The calculation of the porous media model mainly involves the viscous resistance coefficient and the inertial loss coefficient [45,46,47].

The calculation expressions of the viscous resistance coefficient and inertial loss coefficient are as follows:(4)$$C_{1}=\frac{150{\left(1-\varepsilon \right)}^{2}}{D^{2}\varepsilon^{2}}\;$$
(5)$$q=\frac{1}{C_{1}}$$
(6)$$C_{2}=\frac{3.5\left(1-\varepsilon \right)}{D\varepsilon^{2}}$$
where *C*_1_ is the resistance coefficient (m^−2^); *C*_2_ is the inertia loss coefficient (m^−1^); *D* is the pore size of the screen in the capture tube; *q* is the permeability (m^2^) and *ε* is the percentage of the sieving area.

The percentage of sieving area *ε* represents the percentage of the sum of screen pore sizes and the total screen area on the surface of the entire screen.
(7)$$\varepsilon =\frac{D^{2}}{{\left(D+d_{s}\right)}^{2}}\times 100\%$$
where *d_s_* is the screen wire diameter. According to the above equation, the key parameters of the screen can be obtained as shown in Table 4.

#### 3.3.4. Fluent Solver Settings

In the solver, the solution method was set to pressure-based, the velocity formula was the absolute method and the influence of gravity was considered. The Realizable k-epsilon model was chosen as the turbulence model, and standard wall functions were adopted. The coupled method was chosen to shorten the computing time in the case of sufficient computing resources. The differential format of the second-order upwind was chosen to improve the computational accuracy.

## 4. Effects of 3-D Trap Structure on Internal and External Flow Fields

The flow field of a typical section Z = 0 is analyzed below. Figure 4 is the velocity vector diagram at the ambient flow velocity of 20 cm/s and the screen pore size *D* of 0.150 mm. It can be seen that: (1) The external flow field changes as it flows through the capture tube. Part of the water flow decelerates and enters the capture tube, while the rest flows around the tube. (2) After the water enters the capture tube, its flow velocity is relatively uniform. The flow velocity at the settling tube and the funnel-shaped opening above is low since the entire settling tube is closed. The liquid particles in the region collide and rub, consuming lots of energy, which facilitates the sediment particles carried in the water flow in settling by gravity, thereby realizing the capture and collection of sediments. (3) The average flow velocity in the water flow pipe is 1.89 cm/s. When the pore sizes of the screen are the same at the ambient flow velocity of 30 cm/s, the flow field is shown in Figure 5b. When the average flow velocity reaches 2.39 cm/s, the difference between the flow velocity inside and outside the capture tube further increases.

### 4.1. Comparison of the Internal and External Flow Fields of the Capture Tube with Three Screen Pore Sizes

The distributions of the flow field with different screen pore sizes at the same ambient flow velocity were compared. Three velocity vector diagrams for a 30 cm/s flow velocity and three screen pore sizes were analyzed as examples. Figure 5a–c is the internal velocity vector diagrams of the capture tube for the screen pore sizes *D* of 0.300 mm, 0.150 mm and 0.075 mm, respectively. The distribution of the external flow field is relatively similar for different pore sizes. Since there is no significant difference due to the changes in screen pore sizes, so the external flow field distribution is not analyzed. For the same flow velocity and different screen pore sizes, the smaller the screen pore size, the lower the average flow velocity inside the capture tube, and the more concentrated the water flow through the screen is at the center of the screen.

### 4.2. Correspondence of the Flow Velocity Inside and Outside the Capture Tube

During the actual observation and sampling of sediment traps on the seafloor, the structure of the capture tube and the pore size of the screen can influence the ambient flow field. To quantify the effect of screen pore sizes on the flow velocity inside and outside the capture tube, the correspondence between the average flow velocity of the cross-section inside the tube and the ambient flow velocity outside the tube for the three screen pore sizes is summarized in Figure 6.

As shown in the above figure, the three curves have similar trends, and the flow velocities inside and outside the tube are positively correlated. However, the slope of the curves decreases with the increase in the ambient flow velocity outside the tube. When the flow velocity outside the tube increases, the flow velocity inside the tube increases slowly. At the same screen pore size, the influence on the flow velocity is not correspondingly larger with the sharp increase in the flow velocity. When the pore size of screen *D* increases from 0.150 mm to 0.300 mm, the flow velocity in the tube decreases significantly.

Liu (2022) conducted experiments on the flow velocity inside and outside the capture tube in an annular flow tank of 1.8 m in length, 1.1 m in width and 0.13 m in water depth. In the experiment, the velocity of the flow tank was controlled at 10–40 cm/s with a screen pore size *D* = 0.150 mm. By comparing the numerical simulations with the water tank experiment results, a reasonable agreement was achieved between the numerical simulations and the experimental measurements. A slight difference may have been caused by the limitations of the mechanical flow monitoring position chosen and the problem of filter screen blockage in Liu’s experiment [33].

Based on the simulation of flow fields with different screen pore sizes *D* = 0.300 mm, *D* = 0.150 mm and *D* = 0.075 mm and the ambient flow velocity of 15–35 cm/s, the correspondence between the flow velocities inside and outside the capture tube for three screen pore sizes was obtained. A function fitting was performed in Matlab software to establish functional relationships between the flow velocity inside the capture tube *V_in_*, the screen pore size *D*, and the ambient flow velocity outside the tube *V_out_* (Equation (8)). The functional image is shown in Figure 7.
(8)$$V_{in}=-28.77D^{2}-0.001283V_{out}{}^{2}+0.09261D*V_{out}+12.64D+0.09693V_{out}\;-0.9943\;\;\;(R^{2}\;=0.9922)$$

According to Figure 7, the relationship between the flow velocity inside and outside the capture tube can be obtained at a certain screen pore size, which supports establishing the analytical relationship of sediment transport fluxes based on the flow velocity and suspended sand concentration.

## 5. Sediment Capture Efficiency of 3-D Trap

### 5.1. Influence of Changes in Flow Fields on Sediment Particle Motion

The water flow carries the movement of sediment particles, and the movement of sediment has the forms of incipient motion, suspension, sedimentation, etc. In the actual sediment transport process, the water flow carries sediments to flow. The above flow field distribution inside and outside the capture tube obtained by the numerical simulation of the flow field suggests that the flow velocity is reduced due to the capture tube and the screen when passing through the nozzle and inside the capture tube. Part of the sediment group settles after entering the capture tube, and part of the particles bypass the trap with the water flow. Therefore, it is important to calculate the uplift velocity and stop flow velocity of different sediment particle sizes using the equations of sediment uplift velocity and stop flow velocity and to analyze the effect of the changes in flow fields inside and outside the capture tube on the movement of sediment particles.

The critical water flow velocity at which the sediment changes from a moving state to a stationary state is called the stop flow velocity. Through analysis, Sha found that the stop flow velocity of sediment is about 4/5 times the uplift flow velocity. According to the test data, the equation of the uplift flow velocity can be obtained [48] as:(9)$$V_{H}=\frac{4}{5}V_{f}\;$$
(10)Vf=10.5g58γs−γγ58d58ω014RR115 
where *V_H_* is the stop flow velocity of sediment; *V_f_* is the uplift flow velocity of sediment; γ denotes the unit weight of water, taken as 10.0 KN/m^3^; *d* is the sediment particle size (mm); $\gamma_{s}$ refers to the dry unit weight of sediments, taken as 26.5 KN/m^3^; *R*_1_ is the unit hydraulic radius, *R*_1_ = 1 m; *R* denotes the hydraulic radius of the capture tube, *R* = 0.02 m; *g* represents the gravitational acceleration, *g* = 9.8 m/s^2^; and $\omega_{0}$ is the settling speed of sediment particles.

According to studies, when the particle size of sediment particles is larger than 0.03 mm, the flocculation effect is not obvious, and the sedimentation velocity of the sediment group is the same as that of the single-particle sediment. According to the superposition principle of resistance, the equation of sediment particle settling velocity can be derived as follows [49,50,51]:(11)$$\omega_{0}=\sqrt{{\left(13.95\frac{v}{d}\right)}^{2}+1.09\frac{\gamma_{s}-\gamma }{\gamma }gd}-13.95\frac{v}{d}\;$$
where v  is the dynamic viscosity coefficient of water; at 20 °C, v≈0.01 cm^2^/s.

As a general equation for sediment settling velocity, it can meet the requirements of different flow regimes (laminar, transitional and turbulent flows). According to the above equation, the uplift and stop flow velocity curves can be obtained for different sediment particle sizes. Since the starting process of sediment particles is not considered, the two curves divide the motion of particles with different particle sizes into the suspension zone, the push zone, and the stationary or settling zone. The force of the water flow on the particles in the suspension and push zones is greater than the effective gravity in the water, and the particles move with the water flow. In contrast, the force of the water flow on the particles in the stationary or settling zone is smaller than their effective gravity in the water, and the particles undergo a settling motion, as shown in Figure 8.

### 5.2. Capture Efficiency of the Trap

We analyzed the sediment particle capture via the capture tube by combining the sediment movement with the flow field distribution inside the capture tube. The curve diagram of sediment flow velocity (Figure 8) and the diagram of the flow velocity inside and outside the capture tube (Figure 6) were combined into the same figure to obtain the settlement law of sediment particles inside the capture tube. Thus, the accurate capture of sediment particles with different particle sizes was achieved, and the optimal capture efficiency of the sediment trap was obtained.

The range of sediment particle sizes trapped by the capture tube at the screen pore size *D* of 0.3 mm is shown in Figure 9a. According to the motion law of sediment particles and the velocity distribution of the capture tube, at the ambient flow velocity *V_out_* < 17.1 cm/s, only sediment particles with a particle size *d* < 0.3 mm can enter the capture tube; the sediment is in the push or suspension motion state in the capture tube and escapes from the outlet with the water flow, failing to be captured. At the ambient flow velocity *V_out_* > 17.1 cm/s, sediment particles with a particle size *d* > 0.3 mm can enter the capture tube, be intercepted and captured by the screen. At the ambient flow velocity *V_out_
*= 35 cm/s, sediment particles with a particle size *d* < 0.75 mm can enter the capture tube; sediment particles satisfying the particle size in the interval of 0.3 mm < *d* < 0.75 mm can be intercepted and captured by the screen, as shown in the blue shaded area in the figure.

The sediment particle size range trapped by the capture tube at the screen pore size *D* of 0.15 mm is shown in Figure 9b. As shown by the motion law of sediment particles and the velocity distribution of the capture tube, at the ambient flow velocity *V_out_
*< 15.4 cm/s, only the sediment particles *d* < 0.15 mm can enter the capture tube; when the sediment is in the push or suspension motion state in the capture tube, it escapes from the outlet with the water flow without being captured. At the ambient flow velocity *V_out_* > 15.4 cm/s, sediment particles with a particle size *d* > 0.15 mm can enter the capture tube, be intercepted and captured by the screen. At the ambient flow velocity *V_out_
*= 35 cm/s, sediment particles with a particle size *d* < 0.6 mm can enter the capture tube; sediment particles with a particle size within 0.15 mm < *d* < 0.6 mm are intercepted and captured by the screen, as indicated by the green shaded area in the figure.

When the screen pore size *D* is 0.075 mm, the capture range of sediment particle size by the capture tube is shown in Figure 9c. According to the motion law of sediment particles and the velocity distribution of the capture tube, at the ambient flow velocity *V_out_
*< 24.1 cm/s, only the sediment particles with a particle size *d* < 0.075 mm can enter the capture tube; in the push or suspension motion state in the capture tube, the sediment escapes from the outlet with the water flow, failing to be captured. At the ambient flow velocity *V_out_* > 24.1 cm/s, sediment particles with a particle size *d* > 0.075 mm can enter the capture tube, be intercepted and captured by the screen. At the ambient flow velocity *V_out_* = 35 cm/s, sediment particles with a particle size *d* < 0.25 mm can enter the capture tube; sediment particles with a particle size within 0.075 mm < *d* < 0.25 mm are intercepted and captured by the screen, as shown in the yellow shaded area in the figure.

In short, the sediment particle sizes and flow velocity range that can be captured were obtained for different screen pore sizes to accurately capture sediments of specific particle sizes and to classify them by size. The percentage content of sediments with different particle sizes can be obtained by combining in situ water sampling and other methods. According to the percentage of sediments with different particle sizes and the content of sediments with specific particle sizes, the content of other sediment particles of various particle sizes can be derived.

In future indoor or sea trials of the trap, the screen pore size should be appropriately selected under different in situ observation environments. Moreover, the external flow velocity can be obtained by the inversion of the captured sediment particle size to construct the correspondence between the sediment particle size, in-tube flow velocity, and ambient flow velocity.

## 6. Conclusions and Prospects

(1)The flow fields inside and outside the capture tube were numerically simulated to investigate the influence of the capture tube structure, screen pore size and other factors on the on-site flow field. Then, we obtained the functional relationship between the velocity inside the capture tube *V_in_*, the pore size *D* of the filter screen and the ambient flow velocity outside the tube *V_out_*:


$$V_{in}=-28.77D^{2}-0.001283V_{out}{}^{2}+0.09261D\times V_{out}+12.64D+0.09693V_{out}-0.9943$$


(R^2^ = 0.9922). The relationship can provide a theoretical basis for the subsequent inversion of sediment transport fluxes in external ambient flow fields and the establishment of analytical relationships of sediment transport fluxes based on flow velocity and suspended sand concentration.
(2)The relationship between the internal and external flow fields of the trap with three screen pore sizes was obtained. Moreover, by analyzing the movement state of sediment particles, we acquired the sediment particle sizes and flow velocity ranges that can be captured at different screen pore sizes. The appropriate screen pore size was selected when observing different sediment transport environments.(3)The simulation, discussion and analysis in this paper focus on the effects of a single capture tube and different pore sizes of the filter screen on the ambient flow field. The influence of the trap on the flow field also involves the size and shape of the capture tube orifice, the shape of the water flow channel, etc. The research methods in this paper can provide a theoretical basis for the later design of the capture tube structure and the observation of sediment transport fluxes.

## Figures and Tables

**Figure 1 sensors-22-07262-f001:**
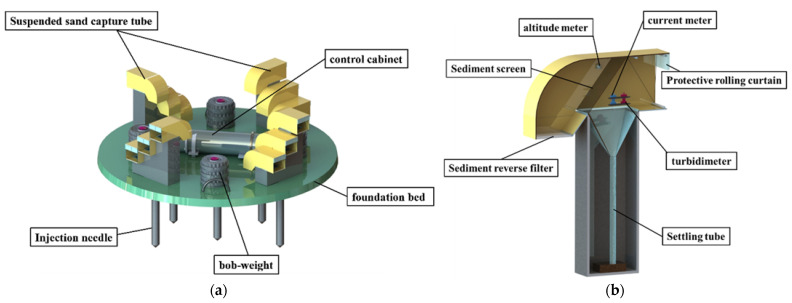
Structure schematic [33]: (**a**) The overall structure of the trap; (**b**) The structure of the capture tube.

**Figure 2 sensors-22-07262-f002:**
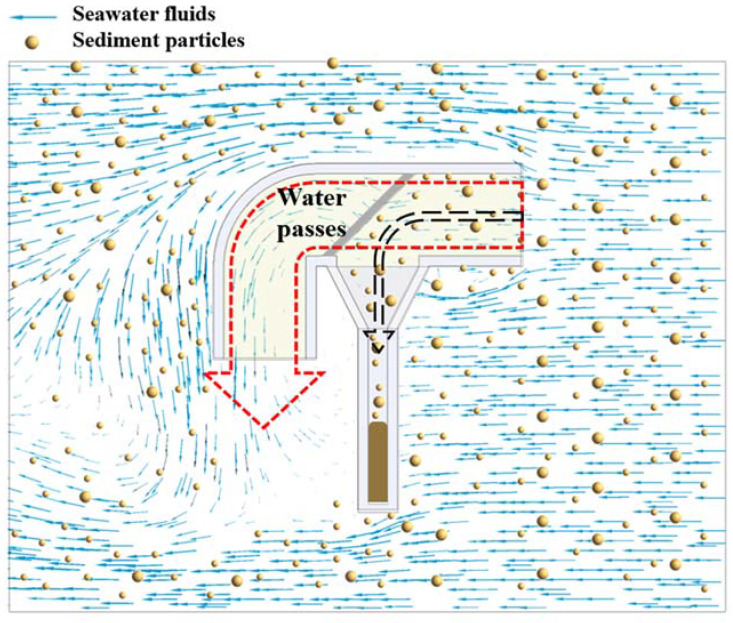
Schematic diagram of the working principle of the capture tube.

**Figure 3 sensors-22-07262-f003:**
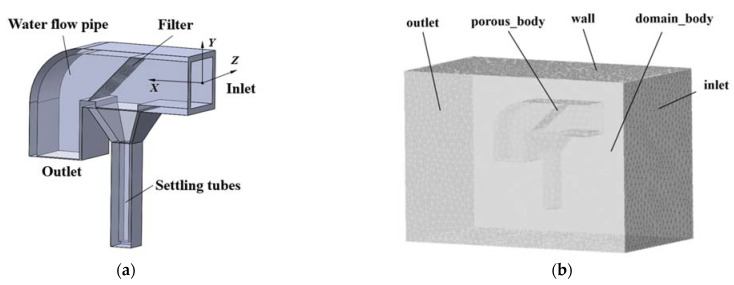
Numerical simulation of the structure: (**a**) 3D model; (**b**) Computational domain mesh model.

**Figure 4 sensors-22-07262-f004:**
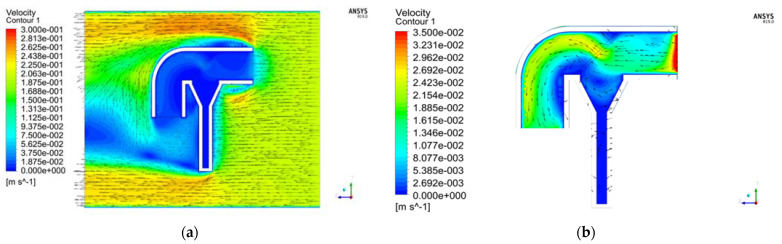
Velocity vector diagram with the ambient flow velocity of 20 cm/s and the screen pore size *D* of 0.150 mm: (**a**) Overall; (**b**) Inside the tube.

**Figure 5 sensors-22-07262-f005:**
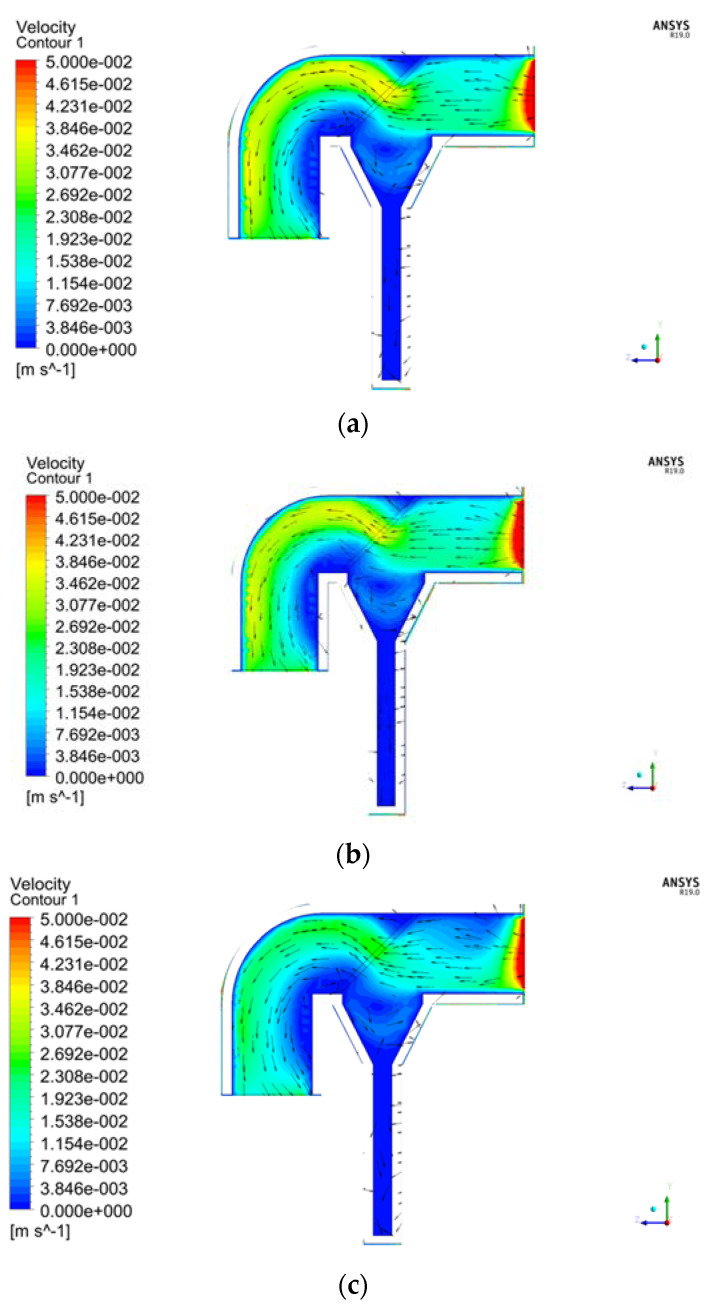
Velocity vector diagram inside the tube for three screen pore sizes at a flow velocity of 30 cm/s: (**a**) *D* = 0.300 mm; (**b**) *D* = 0.150 mm; (**c**) *D* = 0.075 mm.

**Figure 6 sensors-22-07262-f006:**
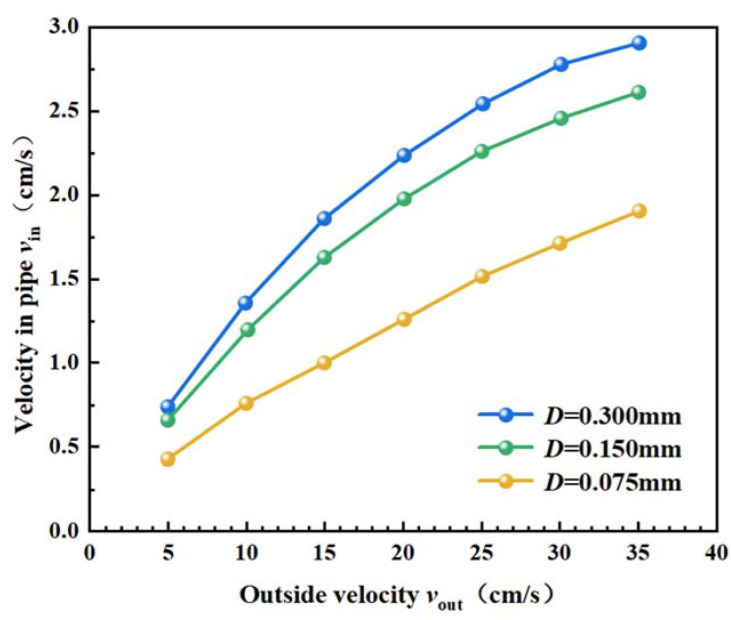
Correspondence of the flow velocity inside and outside the capture tube for three screen pore sizes.

**Figure 7 sensors-22-07262-f007:**
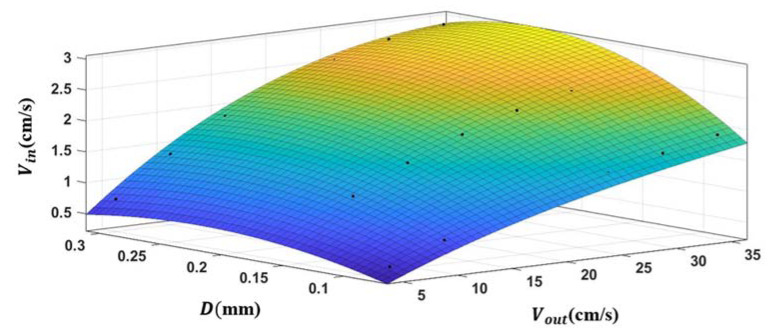
Fitted function image.

**Figure 8 sensors-22-07262-f008:**
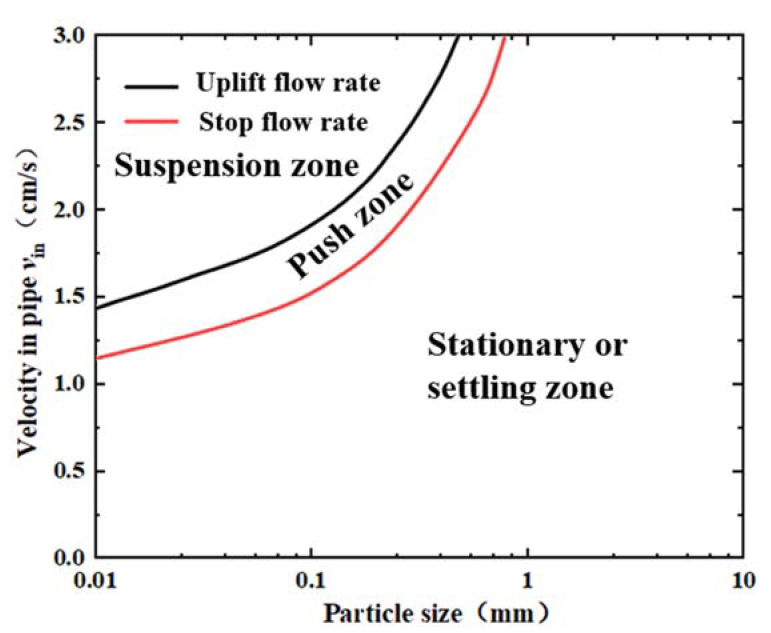
Critical velocity and state of sediment particle motion.

**Figure 9 sensors-22-07262-f009:**
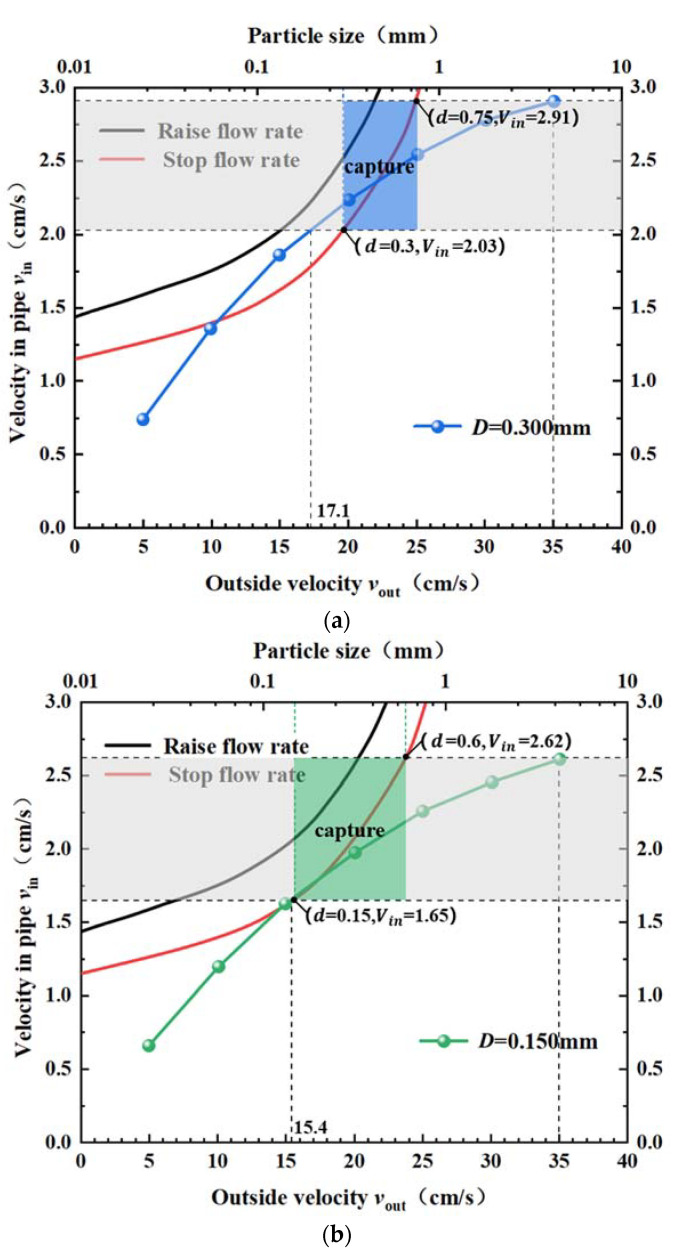
Correspondence between the internal and external flow velocity and critical velocity of sediment: (**a**) *D* = 0.300 mm; (**b**) *D* = 0.150 mm; (**c**) *D* = 0.075 mm.

**Table 1 sensors-22-07262-t001:** Mesh table for different mesh numbers.

Mesh Number (Ten Thousand)	4	10	20	30	40	50
Speed of monitoring surface (cm/s)	0.969	0.989	1.001	1.010	1.014	1.019
Rate of change (%)		2.1	1.2	0.9	0.4	0.5

**Table 2 sensors-22-07262-t002:** Simulation parameters of the continuous phase.

Parameters	Values
Working medium	Seawater (10 °C)
Continuous phase density	1030 kg/m^3^
Continuous phase viscosity	0.001308 kg/(m·s)

**Table 3 sensors-22-07262-t003:** Inlet and outlet simulation parameters.

Parameters	Values	Parameters	Values
Inlet type	Velocity inlet	Outlet type	Pressure outlet
Inlet speed	15–35 cm/s	Outlet pressure	0 MPa

**Table 4 sensors-22-07262-t004:** Main parameters of screens with three pore sizes.

Mesh Number	*D*/(mm)	Screen Thickness/(m)	Screen Wire Diameter /(mm)	*ε*%	*q*/(m^2^)	*C*_1_/(m^−2^)	*C*_2_/(m^−1^)
50	0.300	0.001	0.112	53	2.82 × 10^−10^	3.55 × 10^9^	36,831.30
100	0.150	0.001	0.063	50	7.50 × 10^−10^	1.33 × 10^10^	93,333.33
200	0.075	0.001	0.045	39	2.29 × 10^−10^	4.37 × 10^10^	479,891.2

## Data Availability

Not applicable.

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
