# Peer review of "The Numerical Investigation of the Performance of a Newly Designed Sediment Trap for Horizontal Transport Flux"

_sensors, 2022, doi:10.3390/s22197262_

Round 1
Reviewer 1 Report
The specific points are as follows.
(1) To understand the effect of screen pore sizes on the flow rate inside and outside the capture tube, it is better to add a curve without screen in Fig. 6.
(2) It seems that only one sediment capture tube was simulated in the numerical simulation, in fact, there are many tubes for the designed 3-D trap, the effect of nearby tubes should be discussed or listed as future investigation.
(3) The language of the whole manuscript should be edited. Some sentences are not complete sentences, for example, Line 14, In the abstract “Based on the developed time-series vector observation device for sediment capture and transport process.” ; Line 296, “The water flow carries sediment particles to move, which in the forms of incipient motion, suspension, sedimentation, etc.”
Reviewer 2 Report
See attached file

Reviewer 3 Report
This paper presents an interesting work on numerical simulation of a sediment trap device. The work has practical benefit. However, in its current form, there are some issues that must be addressed before I accept this work.
The writing and presentation are poor. Although the authors are not native speakers, some obvious typos should be avoided. For example:
[1] In author list, “Shaotong Zhang 3,Zihang Fei 2,”, a blank space is lost between two authors.
[2] All variables should be defined when they first appear, also they should be presented in italic form, both in texts and inside figures.
[3] Some paragraphs are too short, and can be merged with others, such as “At present, the commonly used methods for marine sediment observation include in-situ sampling, satellite remote sensing, seated-bottom observation platform and moored sediment traps [13, 14].”
[4] If Figure 1 was published before, then permission to reuse is needed. Also, some labels of Figure 1 such as “foundation xxx” were cut off.
The other issues are:
[1] For the porous media model, authors quoted references [34-44]. However, most of these are old references and there are some recent high-cited ones that should be mentioned as well: MPM simulation of solitary wave run-up on permeable boundaries, Applied Ocean Research, 111, 102602, 2021; SPH modelling of turbulent open channel flow over and within natural gravel beds with rough interfacial boundaries, Advances in Water Resources 140, 103557, 2020; SPH‐based numerical treatment of the interfacial interaction of flow with porous media, International Journal for Numerical Methods in Fluids 92 (4), 219-245, 2020.
[2] Quite a few results were shown in Section 4 and 5. However, there are two issues: they lack validations on sediment motions in any form; and they lack results in 3D dimension but only in the vertical plane. Therefore, it is not clear on the accuracy of your results. Here the authors may not need to do new computations again. As long as they convince the audience their models are accurate, then we can accept all the results.
[3] For all velocity contours such as Figure 4-5, etc, e-003 is used, maybe use 10-3?
[4] Why were Eqs 9 and 10 typed in bold letters? Why were the variables displayed below in different lines? It seems the authors’ writing skills are very poor.
[5] The simulations were all shown in 2D but not in 3D clearly? At least, some qualitative 3D figures can be shown.
[6] Although mesh convergence is done, this is not called critical validation of the model, unless you compare with experimental, field or other numerical results.
[7] For points 5 and 6, I give flexibility to authors to address but the decision will be based on their responses.
Round 2
Reviewer 2 Report
The authors responded satisfactorily to my comments. The manuscript can be accepted in the present form.
Reviewer 3 Report
The authors have made a professional revision and fully demonstrated their scientific working styles. I agree the paper be accepted as is.